# Body Composition of Infants Born with Intrauterine Growth Restriction: A Systematic Review and Meta-Analysis

**DOI:** 10.3390/nu14051085

**Published:** 2022-03-04

**Authors:** Rukman Manapurath, Barsha Gadapani, Luís Pereira-da-Silva

**Affiliations:** 1Maternal and Child Health (Nutrition), Society for Applied Studies, Centre for Health Research and Development, 45-Kalusarai, New Delhi 110016, India; rukman.manapurath@sas.org.in; 2Maternal and Child Health (Implementation Research), Society for Applied Studies, Centre for Health Research and Development, 45-Kalusarai, New Delhi 110016, India; barsha.pathak@sas.org.in; 3Comprehensive Health Research Centre, Medicine of Woman, Childhood and Adolescence, NOVA Medical School|Faculdade de Ciências Médicas, Universidade Nova de Lisboa, Campo dos Mártires da Pátria, Nr 130, 1169-056 Lisbon, Portugal; 4NICU, Hospital Dona Estefânia, Centro Hospitalar Universitário de Lisboa Central, Rua Jacinta Marto, 1169-045 Lisbon, Portugal

**Keywords:** adiposity, body composition, fat-free mass, fetal growth, intrauterine growth restriction, small-for-gestational age

## Abstract

Intrauterine growth restriction (IUGR) may predispose metabolic diseases in later life. Changes in fat-free mass (FFM) and fat mass (FM) may explain this metabolic risk. This review studied the effect of IUGR on body composition in early infancy. Five databases and included studies from all countries published from 2000 until August 2021 were searched. Participants were IUGR or small-for-gestational age (SGA) infants, and the primary outcomes were FFM and FM. Eighteen studies met the inclusion criteria, of which seven were included in the meta-analysis of primary outcomes. Overall, intrauterine growth-restricted and SGA infants were lighter and shorter than normal intrauterine growth and appropriate-for-gestational age infants, respectively, from birth to the latest follow up. They had lower FFM [mean difference −429.19 (*p* = 0.02)] and FM [mean difference −282.9 (*p* < 0.001)]. The issue of whether lower FFM and FM as reasons for future metabolic risk in IUGR infants is intriguing which could be explored in further research with longer follow-up. This review, the first of its kind can be useful for developing nutrition targeted interventions for IUGR infants in future.

## 1. Introduction

According to Barker’s hypothesis, intrauterine growth restriction (IUGR) predisposes to metabolic diseases in later life [1]. Many of the studies used the terms “IUGR” and “small for gestational age” (SGA) synonymously [2]. SGA is commonly defined as those infants whose birth weight is less than the 10th percentile for gestational age or two standard deviations below the population norms on the growth charts. However, the definition does not consider intrauterine growth trajectory and physical characteristics at birth [3,4,5]. An IUGR is a clinical definition and applies to fetal growth deceleration and/or neonates born with clinical features of intrauterine malnourishment, irrespective of their birth weight percentile in relation to gestational age [4]. SGA infants may constitute intrauterine growth restricted as well as constitutionally small infants [3].

Existing evidence suggests both IUGR and SGA have been associated with metabolic complications in later life [6]. Extensive research has shown that increased body fatness or fast postnatal catch-up growth are the reasons for the metabolic risk [7,8]. Additionally, evidence from longitudinal studies also reported that low body fat mass (FM) in early childhood may precede early adiposity rebound which predisposes toward late obesity [9]. There is less evidence on the effect of IUGR on postnatal body composition. This information is important to design nutritional interventions for intrauterine growth-restricted or SGA infants [10,11].

There is a growing body of literature that reports that infants born as growth restricted have a different body composition than those born with normal intrauterine growth [5,12,13]. Existing systematic reviews on body composition of infants born preterm did not distinguish whether the infants suffered IUGR or were born with normal intrauterine growth [14]. Much uncertainty still exists about the effect of fetal growth restriction on the quality of growth in early infancy, based on body composition assessment. Prolonging intrauterine malnutrition into extrauterine life may have specific consequences as undernourishment in the early postnatal months with subsequent rapid catch-up growth is a risk factor for chronic diseases related to nutrition in adult life [15]. The commonly used measurements of body composition are FM to assess body fat, fat-free mass (FFM) as a surrogate of body lean mass, and the derived indices percentage of fat mass (%FM) and fat mass index (FMI) to estimate adiposity.

Hence, the main aim of this systematic review is to investigate the differences in body composition measurements of infants with IUGR compared to infants with normal intrauterine growth during the first postnatal months, both in infants born at term and preterm. Additionally, we intend to determine differences in anthropometry of these infants with respect to weight, length, and head circumference.

## 2. Materials and Methods

### 2.1. Protocol and Registration

The current systematic review and meta-analysis was conducted and reported in accordance with the PRISMA statement checklist and was registered with PROSPERO (Registration number CRD42021272086).

### 2.2. Eligibility Criteria, Information Resources, and Search Strategy

All types of studies published in the English language with body composition assessment using only “direct” and “criterion methods” such as total body water (water/isotope-dilution), total body counting, air displacement plethysmography (ADP), dual-energy X-ray absorptiometry (DEXA), and magnetic resonance imaging (MRI) were considered for this review [16]. We excluded “indirect methods” or “double indirect methods” such as anthropometry and bioelectrical impedance analysis [16]. Studies before 2000 were excluded as they were likely to be of limited relevance due to the latest advancement in methods for body composition assessment. Participant criteria were infants diagnosed at birth as IUGR or SGA who were given standard care. Specifically, IUGR was defined based on fetal growth deceleration and/or clinical features of intrauterine malnourishment at birth, and SGA on birth weight less than the 10th percentile for gestational age or two standard deviations below the population norms on the growth charts [3,4].

We made a comprehensive search between 2000 to 2021, that included original articles published in PubMed (www.ncbi.nlm.nih.gov; accessed on 10 August 2021); Embase (www.embase.com; accessed on 10 August 2021); Web of Science (apps.webofknowledge.com; accessed on 10 August 2021); Scopus (www.scopus.com; accessed on 10 August 2021), and LILACS (LILACS.bvsalud.org; accessed on 10 August 2021). Two review authors (R.M, and B.P.) independently screened all the titles and abstracts identified by the search strategy (Appendix A). Disagreements were resolved by discussion or referring to a third review author (LPdS). Screening and full-text review of the articles were managed using the free web-based software, Rayyan [17]. We extracted data using the modified Cochrane Effective Practice and Organization of Care Group data collection checklist [18].

### 2.3. Data Extraction and Items

The following details were extracted from each and included: study author(s); year of publication; and year in which study was conducted; study design; type, duration, and completeness of follow up (e.g., greater than 80%); country and location of study; fund details; sampling strategy; participant recruitment method; inclusion and exclusion criteria for study; follow-up percentage; pre-specified plan for analysis and dealing missing data and information on the start and end dates of enrolment, gestational age, number of participants (in index and control groups); details on feeding pattern; any comorbidities among enrolled participants, and the method of assessment of body composition. Outcome measures are reported as the mean difference with the associated 95% confidence intervals (CI). Screening and data extraction was managed using the free web-based software, Rayyan (http://rayyan.qcri.org; accessed on 10 November 2021).

### 2.4. Assessment of Quality

The assessment of the risk of bias was conducted to define the methodological quality of the included studies using the Newcastle–Ottawa Scale (NOS). The selection, comparability, and outcome were scored, and the total score was categorized as high risk of bias (total score of 0–2), moderate risk of bias (total score of 3–5), and low risk of bias (total score of 6–8) [19]. Two review authors (B.P. and R.M.) independently assessed the risk of bias [20]. We also ruled out the potential chance of publication bias using egger’s score. The main concern for bias and imprecision of the included studies were the precision and validity of the methods used and the criteria to define IUGR. The decision was taken to do meta-analysis for only those studies that used the criterion method of body composition assessment and using population-based growth charts for defining SGA and matching for gestational age at assessment.

Since we have only pooled studies with the same criteria for defining the study population, no separate assessment of heterogeneity was performed. Additionally, the number of studies at each time point were lesser, hence no sensitivity analysis was conducted.

### 2.5. Data Synthesis

We used the methods described in the Cochrane Handbook for Systematic Reviews of Interventions (Version 6.2, 2021) for data extraction and analysis [20]. The primary analysis was meta-analysis, but a narrative synthesis has been used where studies were unable to pool. The fixed effect meta-analysis (inverse variance method) was used to combine data when study outcomes were similar in magnitude and direction. Studies that used the same method of body composition assessment were pooled together for meta-analysis. In cases of high heterogeneity (defined as I^2^ value greater than 50% [21]), a random effects model, and Restricted Maximum Likelihood Methods (REML) were used [20]. The outcomes for this review were categorized into primary and secondary. The primary outcome measures for the study were body composition measurements: FFM and FM, and the derived indices %FM, percentage of FFM (%FFM), FMI, and FFMI. The study defined IUGR (prenatal diagnosis of fetal growth restriction by ultrasound biometric or Doppler measurements or clinical diagnosis signs of IUGR at birth) and SGA (birth weight less than the 10th percentile of standard growth charts) were studied separately for analysis. The secondary outcomes (weight, length, and head circumference) were pooled irrespective of the study defined IUGR or SGA. Two age periods were considered in the analysis, the neonatal period (within 28 days of life) and postnatal period up to 6 months of age. Studies included infants born preterm and at term. When infants born preterm (36 weeks of gestation or less) reached term-corrected age (37 to 41 weeks’ postmenstrual age), they were analyzed together with infants born at term. For estimates, we used the mean (SD) of each study. If the study has reported a median with a range for the estimate, the Cochrane formula was used to calculate the mean with the standard deviation of the estimate. The combined mean and SD of the group was used for computing the estimate if there were more than one group with fetal restriction [20]. Infants were the unit of analysis. We presented the results graphically using forest plots. Analyses were conducted using Stata 16 (16.1, StataCorp LLC, College Station, TX, USA) [22].

## 3. Results

### 3.1. Study Selection

The PRISMA flowchart for the review process is shown in Figure 1. The searches resulted in 8035 articles, of which 6400 were unique records following the removal of 1635 duplicates. After screening on the title and abstract, 6351 were excluded and 49 were screened on the full text. Eighteen studies were found to be eligible for inclusion.

### 3.2. Study Description and Results

A summary of studies information is provided in Table 1.

### 3.3. Study Characteristics

The 18 included studies with their participants and characteristics are described in Table 1. Common reasons for exclusion included using fewer sensitive methods for body composition analysis such as an anthropometric or bioelectrical impedance analyzer. The list of excluded studies is given in Appendix A.

### 3.4. Settings

The infants included in this review were recruited from neonatal care units in the United States, Italy, United Kingdom, France, Spain, Germany, Sweden, and India.

### 3.5. Method of Assessment

Nine studies used DEXA [13,23,24,26,27,28,31,33,35], eight used air displacement plethysmography [12,29,30,32,34,36,37,38] and one study used MRI [25] for body composition analysis. No other methods were used to assess body composition in the included studies. Those studies with the same criteria for IUGR assessment or defining SGA have been pooled in the meta-analysis. Three studies met the criteria for inclusion but the data for SGA infants were not available separately, the authors were not able to provide data for the same.

### 3.6. Measurements

From the studies included, 12 measured FM and FFM, 14 measured %FM and only 1 study reported LM/FM and %FFM. A total of 9 studies measured body weight, 8 measured length, and 5 measured head circumferences. Length normalized indices FFMI and FMI were reported in only one study.

### 3.7. Participant Characteristics

The studies included a total of 1589 infants with a mean birth weight of 2461.2 g in the index group and 2860.5 g in the control group. Most of the studies included breastfed infants or predominantly breastfed or formula-fed infants as part of the standard care in the neonatal unit.

### 3.8. Types of Studies

Out of 18, 6 studies included infants with IUGR to compare to those without IUGR, and 14 studies compared SGA with AGA infants.

### 3.9. Meta-Analysis (IUGR vs. Normal Intrauterine Growth)

#### 3.9.1. Fat Free Mass (in Grams)

##### Neonatal Period

Three studies based on DEXA reported on FFM of IUGR infants compared to those without IUGR [13,26,33]. Two of them [26,33] reported on term infants within 10 postnatal days (day 3 and day 10) and later follow up at 4 months and 12 months of age (Figure 2a). One study [13] was on preterm infants at term age, and at 6 months post term. All studies reported lower FFM in growth-restricted infants compared to those without IUGR.

##### From 6 Weeks to 6 Months of Age

Two of the aforementioned studies which followed up the infants between 6 weeks and 6 months after term age, also found significantly lower FFM in growth-restricted infants compared to those without IUGR [13,33] (Figure 2b).

#### 3.9.2. Fat Mass (in Grams)

##### Neonatal Period

The three studies based on DEXA reported on FM of IUGR infants compared to those without IUGR [13,26,33]. Two of them [26,33] reported on term infants within 10 postnatal days (day 3 and day 10) and later follow up at 4 months and 12 months of age (Figure 2c). One study [13] was on preterm infants at term age and 6 months post-term. All studies reported lower FM in growth-restricted infants compared to those without IUGR.

##### From 6 Weeks to 6 Months of Age

Two of the aforementioned studies which followed up the infants between 6 weeks and 6 months after term age, also found significantly lower FM in growth-restricted infants compared to those without IUGR [13,33] (Figure 2d).

### 3.10. Meta-Analysis (SGA vs. AGA)

#### 3.10.1. Fat Free Mass (in Grams)

Eight studies reported FFM as an outcome for term infants [23,26,27,28,31,35,36,37]. Two studies that included preterm infants assessed FFM when they reached term age [13,34]. However, we were unable to pool the studies for meta-analysis as the reference charts used to define SGA were different. The time point of body composition assessment varied from 3 to 5 postnatal days for the term infants and 8 postnatal weeks for preterm infants. All these studies reported lower FFM among SGA infants compared to AGA infants.

#### 3.10.2. Fat Mass (in Grams)

##### At Term Age

Two studies based on ADP including preterm infants assessed FM at term age, found significantly lower FM in SGA compared to AGA infants [12,34] (Figure 3).

The results of the secondary outcomes of the study are shown in Appendix A.

### 3.11. Quality of Studies

Most of the studies included were of moderate-to-low risk of bias (see Appendix A for bias assessment). The overall methods (ADP, DEXA, and MRI) used for defining the criteria for IUGR or SGA have been validated and show almost similar sensitivity in previous studies [39].

## 4. Discussion

This review was designed to describe the effect of IUGR on the body composition of infants in the first postnatal months and found that those born with IUGR were having significantly lower FFM and FM compared to infants without IUGR. Body composition of SGA compared to AGA infants followed the same pattern as that of intrauterine growth-restricted infants. The most obvious finding to emerge from this study is that IUGR infants or SGA infants did not catch up in FFM or FM up to 6 months of age.

The benefit of having assessed body composition short term after birth is that it may give more insight into the intrauterine growth quality. Accordingly, this study revealed an early IUGR profile characterized by both low body lean and fat mass. These data may be useful for planning and interpreting long-term follow-up studies, contributing to a better understanding of the mechanisms underlying metabolic programming in these individuals.

Particularly at birth or within 10 postnatal days, we found significantly lower FFM and FM among intrauterine growth-restricted infants or SGA infants. Even though no previous review to date didn’t address this issue, the findings concord with another systematic review on preterm infants, not considering their intrauterine growth [14,40]. Overall, the difference in body composition continues the same pattern at a later follow up in the included studies [12,13,33,34]. In growth-restricted preterm infants, van de Lagemaat et al. [13] found lower FFM at term age but no difference at a later follow up. These results were likely to be related to the higher amount of protein in the feeding regime of IUGR infants. It is important that further work should establish the validity of this finding, as the body composition at birth may determine the future metabolic risk [5,41,42].

SGA infants followed the same pattern as that of intrauterine growth-restricted infants. Interestingly, while both IUGR and SGA infants had lower body mass (sum of FFM and FM) than infants born with normal intrauterine growth.

These findings deserve two observations. First, low FFM rather than high FM may be a reason for metabolic complications in adult life of being born intrauterine growth-restricted or SGA [43,44]. Poor growth during fetal life and infancy may permanently constrain FFM and eventually limit metabolic capacity to tolerate a rich diet [45]. Insulin-like growth factor in fetal skeletal muscle has been studied recently for their role in muscle growth, which is affected in the fetus with IUGR [5,46]. Furthermore, in IUGR preterm infants, an early deficit of FFM is of particular concern as it is associated with brain size and poor neurologic outcomes [47,48]. Second, there is evidence of the impact of lower FM on metabolic diseases in later life. In animal models, it was found that depletion of adipose tissue is related to insulin resistance, which predisposes to chronic cardiometabolic disease [7]. This is a matter of concern in IUGR and SGA infants of our study, presenting a significant FM deficit in early infancy. These results provide further support for the hypothesis that the risk of late obesity exists if an early adiposity rebound occurs during childhood [9].

A recent study that defined SGA based on intergrowth 21st standards, gave valuable insight into the pattern of body composition in low–middle income settings [37]. However, we were unable to include this study in the meta-analysis. In concordance with the findings of high-income settings providing standard neonatal care, this study also showed decreased FFM and FM in SGA infants compared to AGA infants. More evidence from low-income settings is required, where postnatal adverse conditions may impair growth, including for infants born at term [15].

One unanticipated finding from our review was that one of the studies compared IUGR SGA and AGA infants with SGA and AGA infants with normal intrauterine growth, and found significantly lower %FM among IUGR AGA infants but not in SGA infants [26]. This finding should be interpreted with caution as the IUGR was defined at birth based on fetal growth velocity between 22 weeks and birth. Another study has assessed FM gain in preterm infants after term corrected age. Consistent with our finding at term age, IUGR AGA and SGA infants had significantly lower FM than AGA infants with normal intrauterine growth. Three months after term, FM was similar across all groups. These findings cannot be extrapolated to all SGA or IUGR AGA infants receiving routine care, as these groups received slightly higher energy intakes across the study period, which might have caused increased FM accretion [37]. It seems possible that the body compositions of preterm infants may vary based on the average daily energy and protein intake [49].

With respect to postnatal anthropometry, SGA and intrauterine growth-restricted infants were lighter and shorter than AGA and normal intrauterine growth infants, from birth to the latest follow up (Appendix A). Contrary to earlier literature on gestational age-matched SGA and AGA infants, the head circumference did not differ significantly from the AGA group. This data must be interpreted with caution because of the lower sample size for this outcome [50].

Some study limitations need to be acknowledged. First, we were unable to pool some of the studies due to methodological differences like charts used for defining the population, time of assessment of body composition, and the different methods used for body composition assessment as data from different methods are not interchangeable [51]. Second, when infants born preterm reached term corrected age, they were analyzed together with infants born at term. It is described that preterm infants born with normal intrauterine growth, just because of being preterm, are at greater risk of extrauterine growth restriction due to difficulties in providing optimal nutrition in postnatal life [52,53]. Third, the FMI would be preferred to %FM as an indicator of adiposity, since %FM is a ratio with FM included both in the numerator and denominator (as a component of body mass) [51]. Adjusting FM to a measure unrelated to body fat such as body length, defines adiposity better than %FM [51], even for preterm infants [54]. However, only one study comparing SGA with AGA used FMI [12]. More importantly, in many studies length original measurements were not available or information was missing regarding the reliability of length measurements including interobserver variability. Any inaccuracy in length, when it is squared (as in the FMI), magnifies the error of the index, while losing the ability to differentiate overestimation from underestimation [55]. We were unable to account for individual FM and FFM changes with respect to the length in most of the studies included for the meta-analysis due to the unavailability of the length measurement corresponding to the timepoint of body composition measurement or the sample size of the infants who underwent body composition and length were substantially different [13]. In IUGR compared to normal intrauterine growth, only one study addressed %FM in the context of length and found to be positively correlated with increasing length [25]. In future studies assessing body composition of IUGR infants, a greater focus on length either using length normalized indices [56,57] or regression approach to determine the effect of length and weight on FM and FFM [51] could produce interesting findings that may throw more light on the programming hypothesis of excess FM or FFM accretion and the increased risk of metabolic diseases.

To the best of our knowledge, this is the first systematic review and meta-analysis evaluating, how body composition is affected at birth and short term after birth in infants with IUGR as well as SGA compared to those born with normal intrauterine growth, and this is the major strength of the study. This review lays the groundwork for future nutritional and metabolic research planned on IUGR infants. We suggest relating the infant body composition to antenatal ultrasound biometric or Doppler measurements in future studies to get better evidence of the effect of IUGR on body composition and follow them up to at least two years of age.

## 5. Conclusions

This review shows the effect of IUGR on growth and body composition in early infancy. During the early postnatal months, IUGR and SGA infants are lighter and shorter than normal growth infants. Assessing the quality of growth through body composition shows that they have lower FFM and FM. This is a novel finding and may have important implications in the postnatal management of IUGR infants. SGA infants showed a similar pattern to that of IUGR infants. The issue of whether lower FFM and particularly lower adiposity, as reasons for future metabolic risk in IUGR infants is intriguing which could be explored in further research with longer follow up. The findings of this review can be useful for developing nutrition targeted interventions for IUGR infants. More studies with longer follow up and adequate sample size and classification based on standard growth charts are required to arrive at conclusions on body composition.

## Figures and Tables

**Figure 1 nutrients-14-01085-f001:**
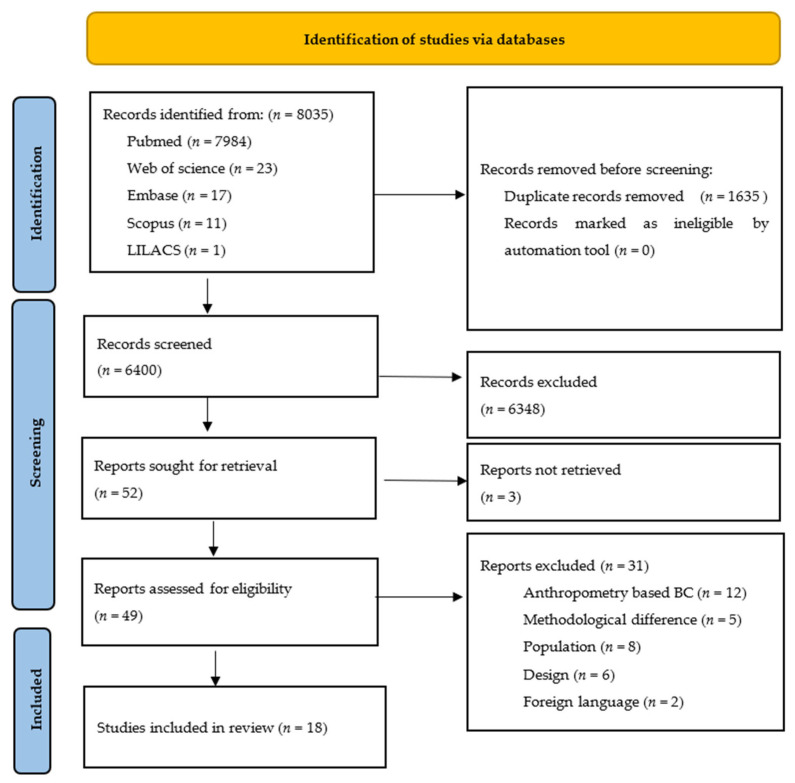
PRISMA screening flowchart.

**Figure 2 nutrients-14-01085-f002:**
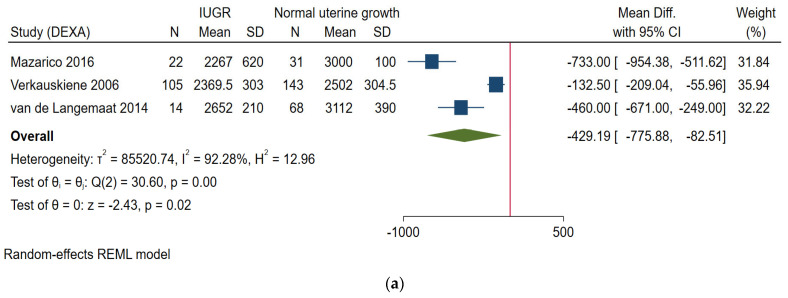
Forest plot showing mean differences between IUGR and normal intrauterine growth infants, in (**a**) FFM-neonatal period; (**b**) FFM-from 6 weeks to 6 months of age; (**c**) FM-neonatal period; (**d**) FM-from 6 weeks to 6 months of age.

**Figure 3 nutrients-14-01085-f003:**
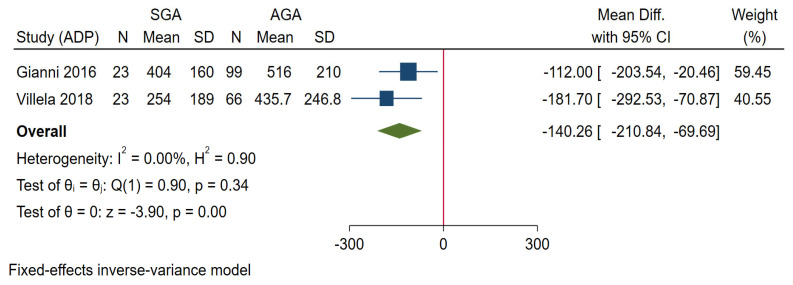
Forest plot showing mean difference FM at term age, between SGA and AGA infants.

**Table 1 nutrients-14-01085-t001:** Description of included studies.

Study	Method	Country	IUGR	Sample Size	Gestational Age (Weeks) of Index Group	Gestational Age (Weeks) of Control Group	Birth Weight (g) Index Group	Birth Weight (g) Control Group	Age at Follow-Up Assessment	Outcomes
					Mean (SD)	Mean (SD)	Mean (SD)	Mean (SD)		
Koo W, 2004 [23]	DEXA	USA	No	90	36.5 (2.6)	35.9 (2.9)	1971 (522)	2454 (634)	3 to 5 days after birth	FFM, FM, FM%
Demarini S, 2006 [24]	DEXA	Italy	Yes	40	36.09/2.3	32.89/3.3	1886 (495)		2 weeks	FFM, FM
Modi N, 2006 [25]	MRI	UK	Yes	29	Term	Term	3.26 (0.43)	2.27 (0.28)	Day 76 weeks	%FM, weight, length
Verkauskiene R, 2007 [26]	DEXA	France	No	248	38.5 (2)	39 (2)	2480 (464)	3227 (449)	Day 3	FFM, %FM, weight, length, head circumference
Ibañez L, 2008 [27]	DEXA	Spain	No	96	39 (0.3)	40 (0.2)	2300 (400)	3400 (400)	14 days	FFM, FM, %FM
Ibañez L, 2010 [28]	DEXA	Spain	No	74	Term	Term	2281 (107)	3409 (92)	15 days, 4 months	FFM, FM, %FM
Moyer-Mileur H, 2009 [29]	ADP	USA	No	43	Term	Term	2535 (246)	3319 (67.6)	Neonatal period	%FM, weight, length, head circumference
Law TL, 2011 [30]	ADP	USA	Yes	87	37.7 (1.5)	38.5 (1.5)	2677 (348)	3273 (591)	Within 7 days	%FM, Weight, length, head circumference
de Zegher F, 2012 [31]	DEXA	Spain	No	174	Term	Term	2900	3900	2 weeks, 4 months	FFM, FM, weight, length, head circumference
Law TL, 2012 [32]	ADP	USA	No	214	30.4 (3.1)	28.4 (3.2)	1148 (421)	1277 (535)	Term age (~10 weeks)	%FM
van de Langemaat *M*, 2014 [13]	DEXA	England	Yes	98	31.1 (1.6)	30.1 (2.0)	1465 (371)	1182 (220)	Term age, 6 months	FFM, FM, %FM, weight, length
Giannì M, 2016 [12]	ADP	Italy	No	122	35.4 (0.77)	35.4 (0.77)	2111 (102)	2468 (331)	5th day of life	FM, %FM, FMI, FFMI, weight, length, head circumference
Mazarico R, 2016 [33]	DEXA	Spain	Yes	48	38.2 (0.2)	37.7 (0.2)	2279 (290)	2208 (210)	10 days, 4 months, 12 months	FM, FFM, weight
Villela L, 2018 [34]	ADP	Germany	No	92	31.8 (1.8)	29.7 (1.5)	1270	1270	Term age, 1, 3, and 5 months	FFM, FM, %FM, %FFM, LM/FM
Schmelzle H, 2007 [35]	DEXA	Germany	No	21	38.2 (2.7)	38.3 (3.0)	2320 (660)	3150 (680)	Within 10 days	%FM, weight, length
Larsson A, 2019 [36]	ADP	Sweden	No	50	38.9 (1.6)	4.1 (1.5)	2499 (209)	4617 (366)	Within 10 days, 4 months	FFM, FM, %FM
Kuriyan R, 2020 [37]	ADP	India	No	153	39 (1.0)	39.5 (1.0)	2700 (100)	3000 (100)	Within 10 days of birth	FFM, FM, %FM
Roggero P, 2011 [38]	ADP	Italy	Yes	195	29.4 (2.2)	29.3(1.8)	1204.8 (253)	1260.8 (198)	Term corrected age, 3 months and 5 months after term	%FM change

## Data Availability

The data presented in this study are available in Modi N, 2006; Verkauskiene R, 2007; Ibañez L, 2008; Ibañez L, 2010; Law TL, 2011; Law TL, 2012; van de Langemaat M, 2014; Giannì M, 2016; Mazarico R, 2016; Villela L, 2018.

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
