# Peer review of "Body Composition of Infants Born with Intrauterine Growth Restriction: A Systematic Review and Meta-Analysis"

_nutrients, 2022, doi:10.3390/nu14051085_

Round 1

Reviewer 1 Report

In this study authors aimed to evaluete the the effect of IUGR on body composition in early infancy and foundet that infants born with IUGR were having significantly lower FFM, FM, and %FM compared to infants without IUGR. Body composition of SGA, compared to AGA infants, followed the same pattern as that of intrauterine growth-restricted infants. This is very interesting topic.

Comments for authors:

First of all I aim surprised from the distrubance regarding the number of studies retrived from different data base such as pubmed , Scopus ,  Embase. Please haw you can explain this fact ; is everything wright in prisma flow chart;

The fact that the infants born preterm (36 weeks of gestation or less) reached term corrected age (37 to 41 weeks’ postmenstrual age), were analyzed together with infants born at term,(It is known that the body composition at term age of infants born preterm is different than that of infants born at termas) well as the fact that the diferent methods use for body composition assessment can be a limitation for this study

Author Response

In this study authors aimed to evaluate the effect of IUGR on body composition in early infancy and found that infants born with IUGR were having significantly lower FFM, FM, and %FM compared to infants without IUGR. Body composition of SGA, compared to AGA infants, followed the same pattern as that of intrauterine growth-restricted infants. This is very interesting topic.

 Comments for authors:

First of all I am surprised from the disturbance regarding the number of studies retrieved from different data base such as PubMed, Scopus, Embase. Please how you can explain this fact; is everything wright in prisma flow chart;

Response: We are thankful for finding this error in the PRISMA flow chart. The necessary edits have been made to the flowchart (Pg 5, Figure 1).

The fact that the infants born preterm (36 weeks of gestation or less) reached term corrected age (37 to 41 weeks’ postmenstrual age), were analyzed together with infants born at term,(It is known that the body composition at term age of infants born preterm is different than that of infants born at term as well as the fact that the different methods use for body composition assessment can be a limitation for this study.

Response: We completely agree with your comment on the limitations of the study. Regarding the combined analysis of preterm infants when they reached term age with term infants, this aspect was already briefly mentioned in the manuscript as a study limitation and in the revised manuscript it is included in a new paragraph gathering the study limitations (Lines 344 – 351).

Reviewer 2 Report

This is an interesting, relevant, well-conducted and well-reported study. However, I have one very important objection. The study reports length of the infants included in the study but these results are not considered in relation to fat mass and fat-free mass. This is a serious limitation. The authors conclusion that fat-free mass (FFM) and especially fat mass (FM) are affected in IUGR and SGA infants may appear quite different if considered in relation to length of the infants. This may have important implications for the interpretation of the study. The authors should handle this problem in the best possible way.

The best alternative is to go through the included studies and check if authors have reported fat-free mass index (FFMI) and fat mass index (FMI). These indexes are often reported in body composition studies. FFMI = fat-free mass (kg)/length (m)2 and FMI = fat mass (kg)/length (m)2. An analysis of these values could then be added to the paper.

A less good alternative is to evaluate average FM and FFM in relation to the average values for length given in the papers. In this way it may be possible to evaluate if affected values for FM and FFM can be explained by effects on length.

If none of these alternatives are possible this should be mentioned in the discussion as a serious limitation and its implications should be discussed in depth.

Author Response

This is an interesting, relevant, well-conducted and well-reported study. However, I have one very important objection. The study reports length of the infants included in the study but these results are not considered in relation to fat mass and fat-free mass. This is a serious limitation. The authors conclusion that fat-free mass (FFM) and especially fat mass (FM) are affected in IUGR and SGA infants may appear quite different if considered in relation to length of the infants. This may have important implications for the interpretation of the study. The authors should handle this problem in the best possible way.

Comments for authors:

The best alternative is to go through the included studies and check if authors have reported fat-free mass index (FFMI) and fat mass index (FMI). These indexes are often reported in body composition studies. FFMI = fat-free mass (kg)/length (m)2 and FMI = fat mass (kg)/length (m)2. An analysis of these values could then be added to the paper.

A less good alternative is to evaluate average FM and FFM in relation to the average values for length given in the papers. In this way it may be possible to evaluate if affected values for FM and FFM can be explained by effects on length.

If none of these alternatives are possible this should be mentioned in the discussion as a serious limitation and its implications should be discussed in depth.

Response: We are thankful for the insightful comments and suggestions. As advised, FMI and FFMI, that have been considered when conceiving the study design is now addressed in the revised Material and Methods section, and results on body composition measurements presented in a new subsection 3.6. Unfortunately, in most studies length original measurements were not available or information was missing regarding the reliability of measurements including interobserver variability. Any inaccuracy in length, when it is squared (as in the FMI), magnifies the error of the index, thus losing the ability to differentiate overestimation from underestimation

Regarding the second alternative, it has not been possible to obtain results based on average FM and FFM in relation to the average values for length, because the sample size of participants who underwent body composition and length measurements were substantially different [van de Lagemaat 2014] and only one study addressed %FM in the context of length [Modi 2006].

As both suggested approaches were not possible or untrustworthy, not adjusting length in body compartments assessment in intrauterine growth-restricted infants has been acknowledged as a major limitation in the revised manuscript as suggested. (Lines 351 – 370)

References

  • van de Lagemaat, M.; Rotteveel, J.; Lafeber, H.; van Weissenbruch, M. Lean mass and fat mass accretion between term age and 6 months post-term in growth-restricted preterm infants. Eur J Clin Nutr. 2014, 68, 1261-1263.
  • Modi, N.; Thomas, E.; Harrington, T.; Uthaya, S.; Doré, C.; Bell, J. Determinants of adiposity during preweaning postnatal growth in appropriately grown and growth-restricted term infants. Pediatric Res 2006, 60, 345-348.

Reviewer 3 Report

Studying the effects of IUGR on body composition in later life is of general interest.

The major limitations of this study are as follows.

First, the observation period is limited to 1 yr after  (and also includes shorter periods of some days after birth). This is a too short period to reach far reaching conclusion. In addition, it is not surprising that within such a short observation period deficits in the nutritional state are maintained.  

Second, there are no data about feeding practices.

Third, although the authors have included studies using reference methods of BCA only there is need to take into account the specific devices as well as the protocols. Strictly spoken, E.g. DXA outcomes do not resemble outcomes obtained by dilution methods etc.

Fourth, I cannot accept that this analyses are restricted to papers published after the year 2000. Contrary to the authors to the author’s view reference methods like DXA, densitometry or isotope dilution are known to be robust for a long time period. These are classical methods. Thus, why do the authors skip these data from their analyses?

Fifth, faced with the graphical presentation of the data in the figures gives me the impression that there is a limited no of high level studies only. If my impression is correct one may question the rationale of this meta-analysis.

Sixth, presenting data on FM and %FM is redundant. In addition, %FM does not provide any meaningful information, since it is included in bw which also is the denominator. In addition, %FM indirectly reflects %FFM questioning again the meaning of this variable.

Seventh, it is a pity that the authors missed the chance to address the major body component (i.e. FFM) where any functional impairment is related to reduced FFM. Thus, the impact of IUGR on the development of FFM has to be addressed too. The authors are referred to the work von Jonanthan Wells (see Adv Nutr. 2013).   

Author Response

Studying the effects of IUGR on body composition in later life is of general interest.

The major limitations of this study are as follows.

First, the observation period is limited to 1 yr after (and also includes shorter periods of some days after birth). This is a too short period to reach far reaching conclusion. In addition, it is not surprising that within such a short observation period deficits in the nutritional state are maintained.  

Response: We absolutely agree with the comment that in such short follow-up period, deficits are expected to be maintained. Nevertheless, we have focused intentionally on a short postnatal period since in this period body compartments are expected to reflect the quality of intrauterine growth. These data may be useful for planning long-term follow-up studies and interpret prolonged deficit of body compartments or fast catch up and contribute to understand the pathways of metabolic programming in the studied population. The rationale for focusing the review on a short postnatal period is now addressed in discussion based on your valuable insight (Lines 283 – 288). The rationale for focusing the review on a short postnatal period is now stated in Discussion (Lines 283 – 288) and addressed in Conclusions (Line 390) based on your valuable insight.

Second, there are no data about feeding practices.

Response: We understand the relevance of feeding practices in this context. Infants included in the analyzed studies were breastfed, predominantly breastfed or formula-fed, as part of the standard care in neonatal units. This has been briefly mentioned in the results section in line number 202-204. No sufficient data were available for a sub group analysis on the effects of different nutrition interventions. Only one study by Vandelagemaat et al. (2014) included infants with slight differences in daily protein and energy intakes.

Third, although the authors have included studies using reference methods of BCA only there is need to take into account the specific devices as well as the protocols. Strictly spoken, E.g. DXA outcomes do not resemble outcomes obtained by dilution methods etc.

Response: We completely agree with the comment. The inclusion criteria has been restricted to studies using direct or criterion methods of body composition analysis (including dilution methods). Accordingly, we have acknowledged as a study limitation that the fact of gathering data in the same analysis from different devices and protocols (Lines 346 – 347). Among these studies on SGA and IUGR infants, dilution methods were not found and hence only DXA and air displacement plethysmography were used.

Fourth, I cannot accept that this analyses are restricted to papers published after the year 2000. Contrary to the authors to the author’s view reference methods like DXA, densitometry or isotope dilution are known to be robust for a long time period. These are classical methods. Thus, why do the authors skip these data from their analyses?

Response: Thank you for the comment. Despite the existence of old studies using robust methods for body composition assessment, we decided to restrict the review after 2000 (21 years) because protocols used before this might be different, particularly definitions of IUGR based on outdated growth charts and body composition data [Wells 2014], absence of complementary data on modern ultrasound fetal biometry or Doppler measurements important to identify SGA infants who suffered IUGR [von Beckerath 2013], and outdated nutritional interventions possibly affecting the body composition [McLeod 2016]. Hence, there was no other intentional exclusion of studies. We understood we were not clear enough and we have added a short statement clarifying this in Results (Line 190).

Fifth, faced with the graphical presentation of the data in the figures gives me the impression that there is a limited no of high level studies only. If my impression is correct one may question the rationale of this meta-analysis.

Response: Thank you for the comment. Concerned in achieving the best possible homogeneity of the studies to pool in the meta-analysis, our inclusion criteria was restricted to “direct” and “criterion methods” for body composition assessment, including total body water (water/isotope-dilution), body counting, air displacement plethysmography, DXA, and MRI. As aforementioned, only studies using DXA and air displacement plethysmography assessing body composition in SGA or IUGR infants were found in the literature search within the last 21 years.

Sixth, presenting data on FM and %FM is redundant. In addition, %FM does not provide any meaningful information, since it is included in bw which also is the denominator. In addition, %FM indirectly reflects %FFM questioning again the meaning of this variable.

Response: Thank you for the insightful comment. Absolute values (weight) of FM and FFM are important but do not provide an insight on the proportion between both compartments. FM in relation of FFM defines adiposity and an individual may have increased FM but a normal adiposity if FFM is proportionally increased. Whole-body fatness assessed by the index FM/FFM would be the best measurement of proportionality [Wells 2005], but representative reference values for this ratio are not available. The best indicator used for adiposity with available reference values is the FMI because it is adjusted to length, a measure unrelated to body fat [Wells 2014, Wells 2020]. However, length original measurements were available only in six of the 18 included studies, and in these we were not confident on the measurement’s reliability since no information was provided on accuracy of measurements’ technique and interobserver variability. Any inaccuracy in length, when this is squared (as in the FMI), magnifies the error of the index while losing the ability to differentiate overestimation from underestimation [Pereira-da-Silva 2018]. As an alternative, we have used the less accurate indicator of adiposity %FM provided in the studies, because this index based on original FM and FFM measurements undertaken by the equipment is much less dependent on the intra- and interobserver variability than that of the length included in FMI. In this context, we used the %FM (as a proportion of body compartments) as a complementary measure to the FM value and we did not consider %FM completely redundant. Of note, 10 included studies reported both these outcomes together. In the revised manuscript, we added as a study limitation the use of %FM as indicator of adiposity, explaining the reason why we did not trust the more accurate FMI in this review. (Lines 350 – 371).

Seventh, it is a pity that the authors missed the chance to address the major body component (i.e., FFM) where any functional impairment is related to reduced FFM. Thus, the impact of IUGR on the development of FFM has to be addressed too. The authors are referred to the work von Jonanthan Wells (see Adv Nutr. 2013).   

Response: Thank you very much for addressing the great article by J. Wells that we have considered a reference article in the revised manuscript. Other two articles by J. Wells (2005, 2020) were also used to support the Discussion. We are also very grateful for the suggestion to highlight the role of FFM. Accordingly, in the revised manuscript the impact of fetal growth on the development of FFM and its metabolic capacity is addressed as well as the important associations of early FFM deficit with poor neurological outcome and brain size (Lines 305 – 312).

References

  • McLeod, G.; Sherriff, J.; Hartmann, P.E.; Nathan, E.; Geddes, D.; Simmer, K. Comparing different methods of human breast milk fortification using measured v. assumed macronutrient composition to target reference growth: a randomised controlled trial. Br J Nutr 2016, 115, 431-439.
  • Pereira-da-Silva L, Virella D. Accurate direct measures are required to validate derived measures. Neonatology. 2018;113(3):266.
  • van de Lagemaat, M.; Rotteveel, J.; Lafeber, H.; van Weissenbruch, M. Lean mass and fat mass accretion between term age and 6 months post-term in growth-restricted preterm infants. Eur J Clin Nutr. 2014, 68, 1261-1263.
  • von Beckerath AK, Kollmann M, Rotky-Fast C, Karpf E, Lang U, Klaritsch P. Perinatal complications and long-term neurodevelopmental outcome of infants with intrauterine growth restriction. Am J Obstet Gynecol. 2013 Feb;208(2):130.e1-6.
  • Wells JC, Victora CG. Indices of whole-body and central adiposity for evaluating the metabolic load of obesity. Int J Obes (Lond). 2005 May;29(5):483-9.
  • Wells JCK, Davies PSW, Fewtrell MS, Cole TJ. Body composition reference charts for UK infants and children aged 6 weeks to 5 years based on measurement of total body water by isotope dilution. Eur J Clin Nutr. 2020 Jan;74(1):141-148.
  • Wells, J.C.K. Toward body composition reference data for infants, children, and adolescents. Adv Nutr 2014, 5, 320S-329S.

Round 2

Reviewer 1 Report

The presented manuscript has been corrected in response to the suggestions. The authors have followed the recommendations of the reviewer. 

Author Response

The presented manuscript has been corrected in response to the suggestions. The authors have followed the recommendations of the reviewer. 

Response: Thank you.

Reviewer 2 Report

The authors have made satisfactory changes to the manuscript. 

Author Response

The authors have made satisfactory changes to the manuscript. 

Response: Thank you.

Reviewer 3 Report

The ms has been improved. There are still two concerns left. 

First, %fat mass is inappropriate  to characterize the nutritional state. This outcome meaningless (i.e., it is biased by bw) and thus should be skipped from the ms. 

Second, in their analyses the authors should not mix up two different methods to assess body composition, i.e., DXA and ADP). These two methods differ in their rationale, outcomes and precision (i.e., with respect to the MDC to assess changes in fat mass). Thus, the data obtained by the different methods should be treated separately.

Author Response

The ms has been improved. There are still two concerns left. 

First, %fat mass is inappropriate to characterize the nutritional state. This outcome meaningless (i.e., it is biased by bw) and thus should be skipped from the ms.

Response: The %FM has been skipped from the manuscript as suggested and changes have been made throughout the text to be consistent with the removal of the %FM importance.

Second, in their analyses the authors should not mix up two different methods to assess body composition, i.e., DXA and ADP). These two methods differ in their rationale, outcomes and precision (i.e., with respect to the MDC to assess changes in fat mass). Thus, the data obtained by the different methods should be treated separately.

Response: As suggested, the data obtained by the different body composition methods have been analyzed separately and the results modified accordingly.